# Research on Real-Time Joint Stiffness Configuration of a Series Parallel Hybrid 7-DOF Humanoid Manipulator in Continuous Motion

Yang Yu *, Shimin Wei *, Haiyan Sheng and Yingkun Zhang

School of Automation, Beijing University of Posts and Telecommunications, No 10, Xitucheng Road, Haidian District, Beijing 100876, China; shenghaiyan@ccbupt.cn (H.S.); zhangyk@bupt.edu.cn (Y.Z.)
* Correspondence: yuyangyy@bupt.edu.cn (Y.Y.); wsmly@bupt.edu.cn (S.W.)

**Abstract:** In this paper, the real-time joint stiffness configuration strategy of a series parallel hybrid 7-DOF (degree of freedom) humanoid manipulator with flexible joints in continuous motion is studied. Firstly, considering the potential human robot accidental collision, combined with the manipulator safety index (MSI) and human body injury thresholds, the motion speed and joint stiffness of the robot are optimized in advance. Secondly, using hyperbolic tangent function for reference, the relationship between joint torques and passive joint deflection angles of the robot is given, which is beneficial for the real-time calculation of joint stiffness and obtain reasonable joint stiffness. Then, the structural model of the selected humanoid manipulator is described, on this basis, the relationship between the joint space stiffness and the Cartesian space stiffness of the humanoid manipulator is analyzed through Jacobian matrix, and the results show that the posture and joint space stiffness of the humanoid manipulator directly affect the Cartesian space stiffness of the humanoid manipulator. Finally, according to whether the humanoid manipulator works in the human-robot interaction environment, the real-time joint stiffness configuration of the humanoid manipulator in continuous motion is simulated and analyzed. The research shows that the humanoid manipulator with flexible joints can adjust the joint stiffness in real-time during continuous motion, and the joint stiffness configuration strategy can effectively improve the safety of human body in human-robot collision. In addition, in application, when the joint space stiffness of the robot is lower, the position accuracy can be improved by trajectory compensation.

**Keywords:** serial parallel hybrid 7-DOF humanoid manipulator; hyperbolic tangent function; human-robot collision; real-time joint stiffness configuration strategy



## 1. Introduction

The research of humanoid manipulators is important in the field of robotics. Compared with general humanoid manipulators, the humanoid manipulator with flexible joints is more anthropomorphic and has important research value [1–3]. A key problem within the research of the humanoid manipulator with flexible joints concerns the configuration of the stiffness of each joint in real-time during the continuous motion of the humanoid manipulator.

At present, the main research object within humanoid manipulators is mostly the serial redundant manipulator [4–7]. Although the serial redundant manipulator can accomplish many complex tasks flexibly, its mechanism is still very different from that of the human arm. Research of human arm movements has confirmed that the mechanism of the human arm is a series parallel hybrid mechanisms. The series of parallel hybrid mechanisms combines the advantages of the series mechanism and the parallel mechanism, and has large workspace and high dynamic performance [8–10]. The SEA (Series Elastic Actuators) was proposed by Pratt et al. in 1995 [11]. Since then, many researchers have begun to study variable stiffness actuators with a view to apply this to the joints of robots. Nowadays, the

typical variable stiffness actuators are VSA (variable stiffness joint actuator) [12], AMASC (actuator with mechanically adjustable series compliance) [13], pVSJ (passive variable stiffness joint) [14], etc. [15–18]. Although research of variable stiffness joint actuators is increasing, research on robots with flexible joints is not yet at a mature stage.

Many researchers have laid the foundation for the research of humanoid manipulator based on flexible joints. It is known that the human arm has good joint flexibility, redundant mechanism, and surface contact flexibility, so that it can flexibly and safely complete many complex tasks. Assuming that the humanoid manipulator is a serial parallel hybrid 7-DOF manipulator, and the stiffness of each joint can be changed continuously, compared with the serial redundant manipulator and the manipulator of the rigid joint, the humanoid manipulator and the human arm have more in common. The joint flexibility of the human arm is adjusted passively according to its' needs, while the joint flexibility of the humanoid manipulator in continuous motion can be configured actively according to the working environment.

The working environment of a robot can be roughly divided into the environment of no interference and the environment of human robot interaction. In the environment of no interference, there is no need to worry about the harm of robot movement to human bodies; in the environment of human robot interaction, it is very important to ensure the safety of the human body. In 2016, the international organization for standardization (ISO) issued the ISO/TS 15,066 technical standard, which provides technical guidance for operators to ensure their safety when working with robots [19,20]. In the human-robot interaction environment, it is very likely that there will be accidental collisions between robots and humans. Many researchers have studied how to avoid these human-robot collisions [21,22], which is important, but it is equally important to reduce injuries sustained to the human body in the event of accidental collision. The flexible joint manipulator can adjust the stiffness of each joint in real-time to meet environmental requirements. Therefore, it is necessary to study how we might reduce the collision force by adjusting the joint stiffness of the humanoid manipulator.

Michael Melia et al. conducted special research on collisions between cooperative robots and human bodies, and obtained the threshold of pain in various parts of the human body when subjected to force and pressure [23,24]. According to this research, when the human and robot collide accidentally, the force on the human body should be lower than the pain threshold to ensure that the human body is safe. In addition, Antonio Bicchi et al. studied the collision problem between a single joint flexible robot and human head, and adopted HIC (Head Injury Criterion) as the injury index of human-robot collision [25–27]. The smaller the HIC is, the safer the human body is. On this basis, Ki Hong Kim and others proposed MSI to predict the injury of human-robot collision [28]. The above research can provide theoretical support for the evaluation of human-robot collision injury. It was found that the velocity of the robot and the stiffness of the human robot contact surface both play an important role in the safety of human-robot collisions. Therefore, in the human-robot interaction environment, we can use the above research for reference to predict and evaluate the safety of human-robot collisions, and establish the joint stiffness configuration strategy to obtain the real-time joint stiffness of the humanoid manipulator in continuous motion, so as to ensure the safety of the human body.

In this paper, the real-time joint stiffness configuration strategy of a serial parallel hybrid 7-DOF humanoid manipulator is studied. In Section 2, the real-time joint stiffness configuration strategy of the humanoid manipulator is established and the structural model of the selected humanoid manipulator is described. In Section 3, the relationship between the joint space stiffness and the Cartesian space stiffness of the humanoid manipulator is analyzed. In Section 4, the joint stiffness real-time configuration of the humanoid manipulator in the environment of robot working alone and human robot interaction is simulated and analyzed, and the feasibility of the proposed method is verified. Finally, the summary of the research work and the prospect of the future work are given.

## 2. Methods

### 2.1. Real-Time Joint Stiffness Configuration Strategy

The working environment of the robot can be divided into two kinds: working alone away from human bodies and working in the human-robot interaction environment. When the robot works far away from the human body, its movement will not cause subsequent harm, so the accuracy of the robot should be guaranteed when the robot performs tasks such as grasping or placing. When the robot works in the human-robot interaction environment, there is a risk of accidental collision with the human body, so it is necessary to take prevention and protection strategies to prevent the human body from being injured.

According to the research of Michael Melia et al. [23,24] on the collision data between cooperative robot and human bodies, the pain threshold of each part of the human body is about 150 N. When the human body feels pain, the energy transfer during human-robot collision [19,20] is as follows:

$$E = \frac{F_c^2}{2K} = \frac{1}{2} m_T v_R^2 \tag{1}$$

where, $F_c$ is the force on the human body in human-robot collision; $K$ is the stiffness of the human robot contact surface; $m_T$ is the converted mass of the whole system in human-robot collision; $v_R$ is the relative motion speed of the human and the robot, and the unit is m/s.

$$m_T = \left( \frac{1}{m_H} + \frac{1}{m_R} \right)^{-1} \tag{2}$$

where, $m_H$ is the effective mass of human body, $m_R$ is the effective mass of the humanoid manipulator.

Ki Hong Kim et al. [28] and others put forward the manipulator safety index (MSI), which is used to predict the injury of human-robot collision. The Equation is as follows:

$$\mathrm{MSI} = \Delta T \left[ \frac{2}{g\Delta T} \left( \frac{m_R}{m_R + m_H} \right) v_R A \right]^{2.5} \tag{3}$$

where

$$A = \begin{cases} \sin(w_n \frac{\Delta T}{2}) & \text{if} \Delta T < \frac{T_c}{2} \\ 1 & \text{if} \Delta T \geq \frac{T_c}{2} \end{cases} \tag{4}$$

In addition, $\Delta T$ is the time interval, and $\Delta T$ is fixed as either 15 or 36 ms, and $T_c$ is the duration of the collision, $T_c = 2\pi/w_n$; $A$ and $w_n$ are constants, and $w_n = \sqrt{\frac{m_R + m_H}{m_R m_H} K}$; $K$ is the same as in Equation (1).

It can be seen that in human-robot collision, whether considering the pain threshold of the human body or the manipulator safety index, the safety of human body is related to the comprehensive surface contact stiffness $K$ and the relative velocity $v_R$ of human and robot. The larger the comprehensive surface contact stiffness $K$ and the relative velocity $v_R$, the greater the force of human-robot collision, and vice versa. Similarly, we know that the larger the comprehensive surface contact stiffness $K$ and the relative velocity $v_R$ in human-robot collision, the shorter the collision time. In order to reduce human injury in human-robot collision, it is necessary to increase the buffer time in collision. Therefore, the smaller the comprehensive surface contact stiffness $K$ and the relative velocity $v_R$ are, the safer the human body is.

Thus, in the continuous motion of the robot, we can protect the human body from being injured in the human-robot collision by optimizing the comprehensive surface contact stiffness $K$ and the relative velocity $v_R$. In this case, both conditions are satisfied at the same time, that is, the force of the human-robot collision is not higher than the human body

injury thresholds, and the safety index MSI is less than $\text{MSI}_{\max}$. The conditional equation is as follows:

$$\begin{cases} F_c = \sqrt{K m_T v_R^2} \leq F_{c\max} \\ \text{MSI} \leq \text{MSI}_{\max} \end{cases} \tag{5}$$

where $F_{c\max}$ is the maximum force that human can bear when human body is not injured in human-robot collision, $F_{c\max} = 150\text{N}$; $\text{MSI}_{\max}$ is the maximum of the safety index, and $\text{MSI}_{\max} = 10$.

For the robot with flexible joints, the joint space stiffness of the robot will affect the comprehensive surface contact stiffness in human-robot collision. Therefore, by adjusting the joint space stiffness of the robot in real-time, the safety of human-robot collision can be improved in the human-robot interaction environment. The joint space stiffness of the robot is related to the joint torque and the passive joint deflection angle. In order to maintain the motion stability of the robot, the closer the joints are to the base coordinate system, a greater the stiffness of joints is required. And when the joint torque is too large, the passive joint deflection angle should be controllable.

According to the above requirements, referring to the hyperbolic tangent function, the relation equation between the joint torque $\boldsymbol{\tau}$ and the passive joint deflection angle $\Delta\boldsymbol{\theta}$ of the robot is proposed as follows:

$$\Delta\boldsymbol{\theta} = \boldsymbol{b}\tanh(\boldsymbol{\tau}/\boldsymbol{a}) \tag{6}$$

where, for 7-DOF humanoid manipulator with flexible joints, $\boldsymbol{a}$ and $\boldsymbol{b}$ are $7 \times 1$ parameter matrices respectively, which can be determined according to the working environment of the humanoid manipulator; the joint torque $\boldsymbol{\tau}$ and the passive joint deflection angle $\Delta\boldsymbol{\theta}$ are $7 \times 1$ matrices respectively.

The relationship curves between the joint torque and the passive joint deflection angle of the humanoid manipulator are S-shaped curve. For example, we assign values to $\boldsymbol{a}$ and $\boldsymbol{b}$, if $\begin{cases} \boldsymbol{a} = \left[\frac{50}{\pi}, \frac{45}{\pi}, \frac{40}{\pi}, \frac{35}{\pi}, \frac{55}{\pi}, \frac{60}{\pi}, \frac{65}{\pi}\right]^T \\ \boldsymbol{b} = \left[5 \times 10^{-4}, 4.5 \times 10^{-4}, 4 \times 10^{-4}, 3.5 \times 10^{-4}, 5.5 \times 10^{-4}, 6 \times 10^{-4}, 6.5 \times 10^{-4}\right]^T \end{cases}$, then the relationship curve between the joint torque and the passive joint deflection angle of the humanoid manipulator is shown in Figure 1.

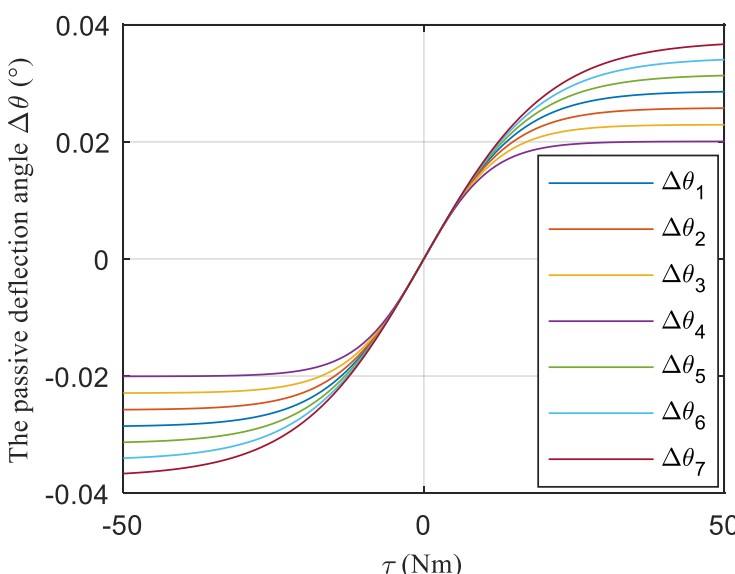

**Figure 1.** The relationship curve between the joint torque and the passive joint deflection angle.

It can be seen from Figure 1 that according to the given parameter matrices $\boldsymbol{a}$ and $\boldsymbol{b}$, no matter what the joint torque of the humanoid manipulator is, the passive joint

deflection angle will always be limited in a certain range. When the joint torque is small, the relationship between the joint torque and the passive joint deflection angle is approximately linear, in this case, the joint space stiffness is basically constant. When the joint torque is large, the passive deflection angle of the joint has little change with the increase of the joint torque, and the joint space stiffness gradually increases. The relationship curve meets the motion stability requirements of the humanoid manipulator.

In addition, the joint torque $\tau$ is calculated as follows:

$$\tau = J^T F \tag{7}$$

where $J^T$ is the force Jacobian matrix and $F$ is the force/torque on the end effector of the robot.

According to Equations (6) and (7), the calculation equation of the joint space stiffness of the humanoid manipulator can be obtained.

$$K_{\theta_i} = \frac{\tau_i}{\Delta \theta_i} \quad (i = 1, \cdots 7) \tag{8}$$

Then the joint stiffness matrix of the humanoid manipulator is as follows:

$$K_\theta = diag\left(K_{\theta_1}, K_{\theta_2}, K_{\theta_3}, K_{\theta_4}, K_{\theta_5}, K_{\theta_6}, K_{\theta_7}\right) \tag{9}$$

Based on the above method, if the humanoid manipulator works close to a human, considering the safety of human-robot collision, the flow chart of joint stiffness configuration strategy is shown in Figure 2.

As shown in Figure 2, in the joint stiffness configuration strategy, we optimize the motion planning time $T_m$ of the humanoid manipulator through the estimated collision force $F_c$, so as to obtain the reasonable motion speed $v_R$ of the humanoid manipulator. The comprehensive surface contact stiffness $K$ between the humanoid manipulator and human body is optimized by the safety index MSI. The motion planning time $T_m$ and constant $A$ in the motion of the humanoid manipulator are given on the condition that the estimated collision force $F_c$ and the safety index MSI are met at the same time. Then the real-time joint stiffness in the motion of the humanoid manipulator is obtained according to Equations (6)–(8).

### 2.2. Structural Model of the Humanoid Manipulator

In order to study the real-time joint stiffness configuration strategy of flexible joint robots in continuous motion, a series parallel hybrid 7-DOF humanoid manipulator model with variable stiffness joints is proposed to verify the feasibility of the proposed method through simulation analysis. Compared with the general series of commercially available humanoid robots, the humanoid manipulator model has better anthropomorphic characteristics in configuration.

The humanoid manipulator consists of a 3-DOF shoulder joint, 2-DOF elbow joint, and 2-DOF wrist joint. The shoulder joint is composed of a 1-DOF revolute joint and an orthogonal 2-DOF parallel mechanism; elbow joint and wrist joint are respectively composed of an orthogonal 2-DOF parallel mechanism. The three joints are connected in series. The joints of the humanoid manipulator are flexible, and the joint stiffness can be changed continuously. The range of joint stiffness is $[1000, \infty]$, and the unit is Nm/rad. The structural model and the mechanism diagram of the humanoid manipulator are shown in Figure 3.

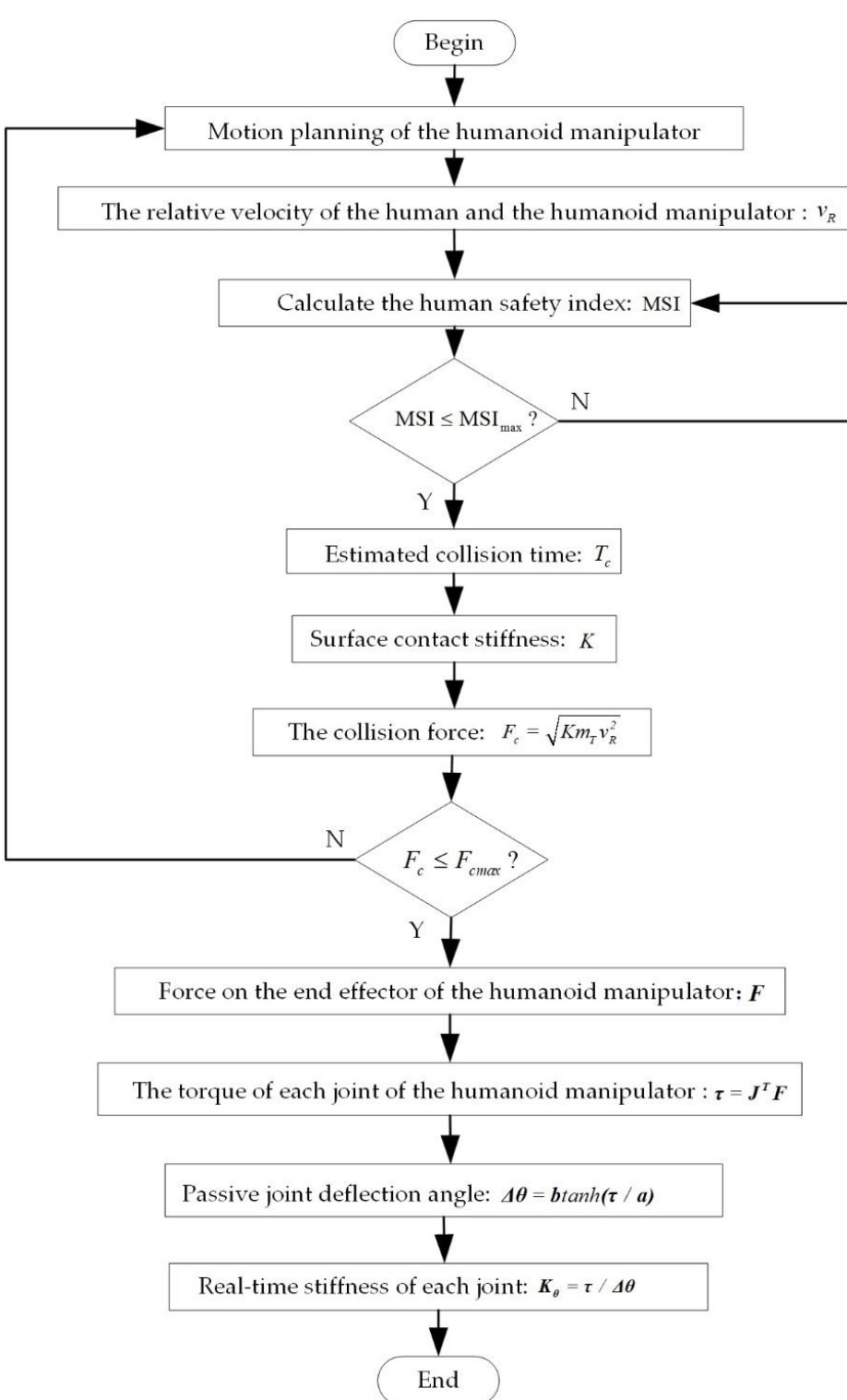

**Figure 2.** The flow chart of joint stiffness configuration strategy.

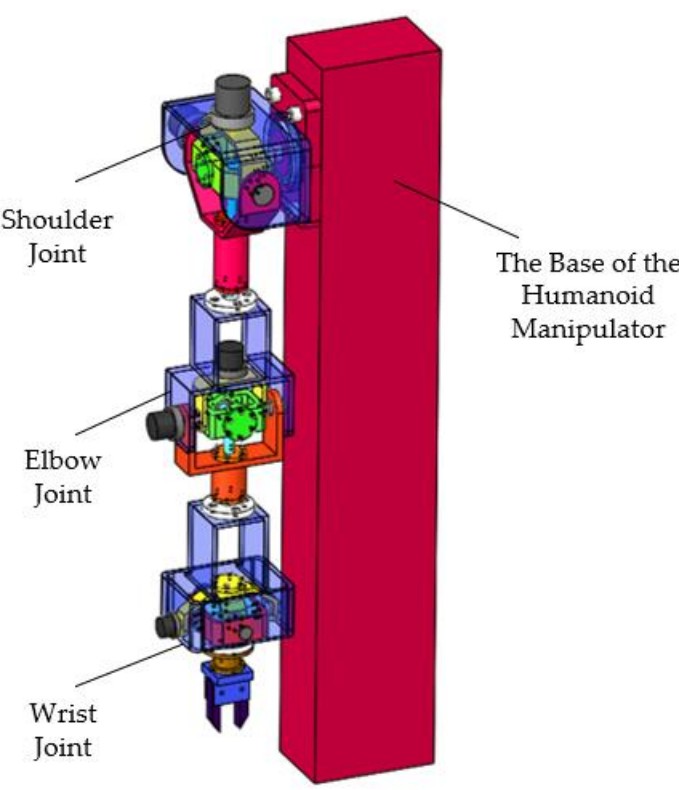

**Figure 3.** The structural model of the humanoid manipulator.

The relevant structure parameters and workspace of the humanoid manipulator are shown in Appendix A.

### 3. The Joint Space Stiffness and the Cartesian Stiffness

The relationship between the joint space stiffness and the Cartesian stiffness [29,30] of the humanoid manipulator is as follows:

$$K_p = J^{-T} K_\theta J^{-1} \tag{10}$$

where $K_p$ is the Cartesian stiffness matrix of $6 \times 6$; $K_\theta$ is the joint space stiffness diagonal matrix of $7 \times 7$, and each diagonal term represents the stiffness of the corresponding joint; $J$ is the Jacobian matrix of $6 \times 7$, which is derived from the forward kinematics of the humanoid manipulator.

Because the Cartesian stiffness matrix is not a diagonal matrix, it is difficult to directly estimate the Cartesian space stiffness of the humanoid manipulator through this matrix. Therefore, the generalized displacement of the end effector of the humanoid manipulator is used to evaluate the Cartesian space stiffness.

$$\Delta p = CF \tag{11}$$

where $\Delta p$ is the generalized displacement of the end effector of the humanoid manipulator, $\Delta p = \left[\Delta x, \Delta y, \Delta z, \Delta\theta_x, \Delta\theta_y, \Delta\theta_z\right]^T$; $F$ is the force/torque on the end effector of the humanoid manipulator, $F = \left[F_x, F_y, F_z, T_x, T_y, T_z\right]^T$; $C$ is the compliance matrix of the humanoid manipulator.

$$C = K_p^{-1} = J K_\theta^{-1} J^T \tag{12}$$

Assuming that the gravitational load is 1 kg, and the coordinate of the center of gravity of the load relative to the coordinate system of the end effector is $(30, 50, -40)$, the unit is millimeter. The relevant calculation equations of the force/torque of gravitational load acting on the end effector are given in Appendix B. The joint angle of the series parallel

hybrid 7-DOF humanoid manipulator is $q = \left[ -\frac{\pi}{3}, -\frac{\pi}{6}, \frac{\pi}{18}, -\frac{7\pi}{18}, \frac{\pi}{36}, -\frac{\pi}{36}, -\frac{\pi}{9} \right]^T$. Three kinds of joint space stiffness of the humanoid manipulator are given as follows:

$$\begin{cases} \boldsymbol{K_{\theta l}} = diag\left[10^4, 10^4, 10^4, 10^4, 10^4, 10^4, 10^4\right] \\ \boldsymbol{K_{\theta m}} = diag\left[5 \times 10^4, 5 \times 10^4, 5 \times 10^4, 5 \times 10^4, 5 \times 10^4, 5 \times 10^4, 5 \times 10^4\right] \\ \boldsymbol{K_{\theta h}} = diag\left[10^5, 10^5, 10^5, 10^5, 10^5, 10^5, 10^5\right] \end{cases},$$

the unit is Nm/rad.

　　Then the Cartesian space stiffness of the humanoid manipulator with different joint space stiffness is compared by comparing the generalized displacement of the end effector of the humanoid manipulator, as shown in Figure 4.

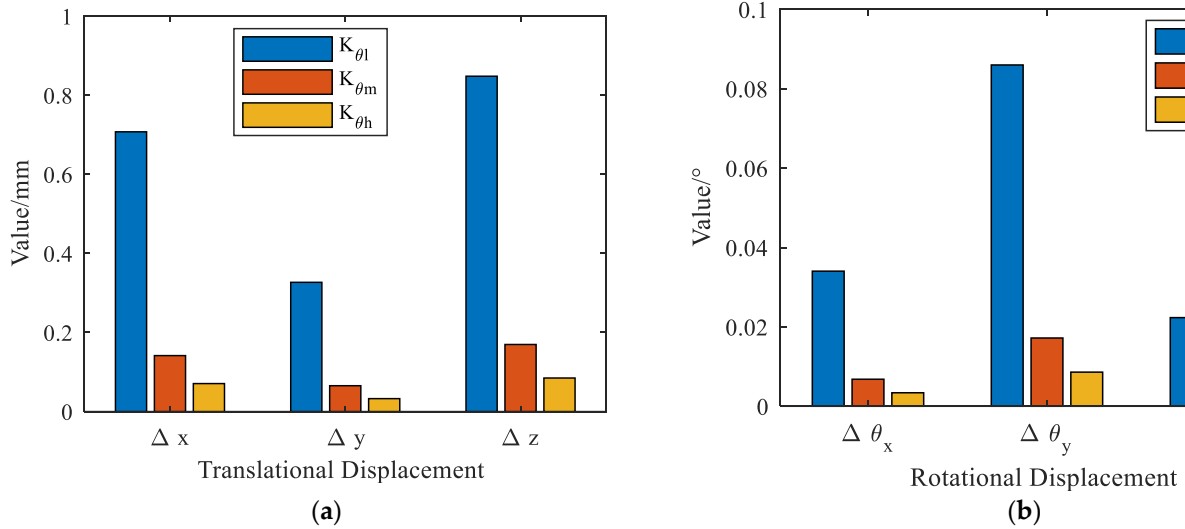

**Figure 4.** The generalized displacements of the end effector of the humanoid manipulator with different joint space stiffness. (**a**) Translational displacements of the end effector; (**b**) Rotational displacements of the end effector.

　　It can be seen from Figure 4 that when the humanoid manipulator has the same posture and load, the change of the joint space stiffness will affect the generalized displacement of the end effector, that is, the Cartesian space stiffness of the humanoid manipulator is directly related to the joint space stiffness of the humanoid manipulator. The smaller the joint space stiffness is, the better the overall flexibility of the humanoid manipulator. In this case, the generalized displacement of the end effector is larger when the same force/torque is applied to the end effector, and vice versa. Therefore, in the human robot interaction, the safety of human robot interaction may be improved by adjusting the joint space stiffness of the humanoid manipulator in real-time.

　　Assuming that the gravitational load on the end effector of the humanoid manipulator is the same as above, the joint space stiffness of the series parallel hybrid 7-DOF humanoid manipulator is $\boldsymbol{K_\theta} = diag\left[5 \times 10^4, 5 \times 10^4, 5 \times 10^4, 5 \times 10^4, 5 \times 10^4, 5 \times 10^4, 5 \times 10^4\right]$, the unit is Nm/rad. Because the humanoid manipulator is redundant, there will be multiple inverse kinematics solutions for the same position and pose of the end effector. We select three groups of different joint output angles, as shown below.

$$\begin{cases} \boldsymbol{q_1} = [-1.0472, -0.5236, 0.1745, -1.2217, 0.0873, -0.0873, -0.3491]^T \\ \boldsymbol{q_2} = [-1.1707, -0.7568, 0.8119, -1.2217, -0.8506, -0.3939, -0.4316]^T \\ \boldsymbol{q_3} = [-1.1707, -0.0581, -0.6549, -1.2217, 1.2775, 0.4518, -0.6507]^T \end{cases},$$

the unit is rad.

　　Based on the above, the generalized displacements of the end effector of the humanoid manipulators with different joint output angles are compared, as shown in Figure 5.

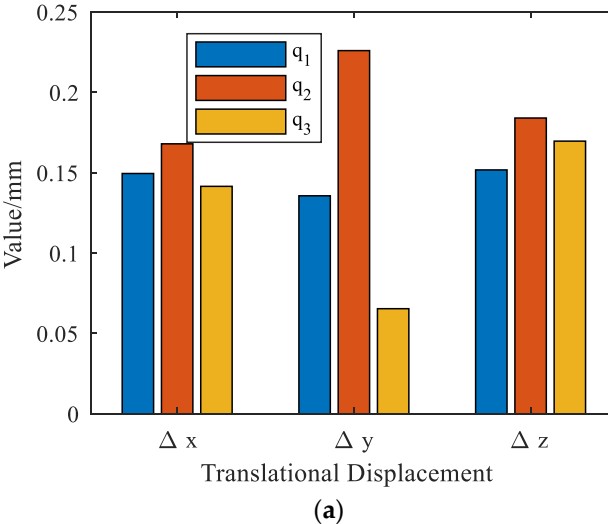

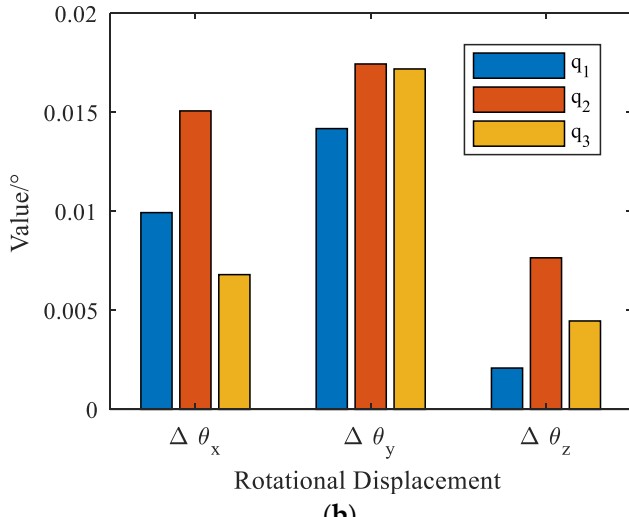

**Figure 5.** The generalized displacements of the end effector of the humanoid manipulators with different joint output angles. (**a**) Translational displacements of the end effector; (**b**) Rotational displacements of the end effector.

It can be seen from Figure 5 that when the humanoid manipulator has the same joint space stiffness and the same force/torque is applied to the end effector, if the joint output angles of the humanoid manipulator are different, the Cartesian space stiffness of the humanoid manipulator is also different. Perhaps the desired Cartesian space stiffness can be obtained by selecting the appropriate inverse kinematics solution of the humanoid manipulator.

## 4. Simulation Analysis

Since the joint space stiffness of the humanoid manipulator can be adjusted in real-time, the joint space stiffness will directly affect the Cartesian space stiffness of the humanoid manipulator. Therefore, according to the above proposed real-time joint stiffness configuration strategy, we can adjust the joint space stiffness of the humanoid manipulator in real-time according to the working environment, so as to ensure that the humanoid manipulator can work normally and will not cause harm to the human body in the event of accidental collision.

### 4.1. In the Environment where the Robot Works Alone

When the humanoid manipulator works alone, in order to ensure the operation accuracy, we hope that the generalized displacements of the end effector of the humanoid manipulator are within a certain range, and when the end effector is suddenly subjected to a large force/torque, the generalized displacements of the end effector will not be so large that it will cause the humanoid manipulator to vibrate. Therefore, it is necessary to adjust joint space stiffness according to the real-time joint torque of the humanoid manipulator, to control the Cartesian space stiffness in real-time.

Assuming that the gravitational load is 3 kg, and the coordinate of the center of gravity of the load relative to the coordinate system of the end effector is $(30, 50, -40)$, the unit is millimeter. The starting point and end point of the end effector of the humanoid manipulator are $P_A = (300, -500, 700, 30, -100, 160)$ and $P_B = (900, 100, 300, 40, -80, 120)$, and the units of position and Euler angle are millimeter and degree, respectively. The path planning is based on quintic polynomial, and the running time is set to 5 s. The joint torque $\boldsymbol{\tau}$ and the passive joint deflection angle $\Delta\boldsymbol{\theta}$ of each joint conforms to the hyperbolic tangent relation equation, and $a_i = \frac{50}{\pi}$, $b_i = 5 \times 10^{-5}$, $i = 1 \cdots 7$.

The real-time situation of force/torque on the end effector, the joint space stiffness, etc., of the humanoid manipulator are shown in Figures 6–8 respectively. The joint angle and

real-time trajectory of the end effector of the humanoid manipulator in real-time motion are shown in Appendix C.

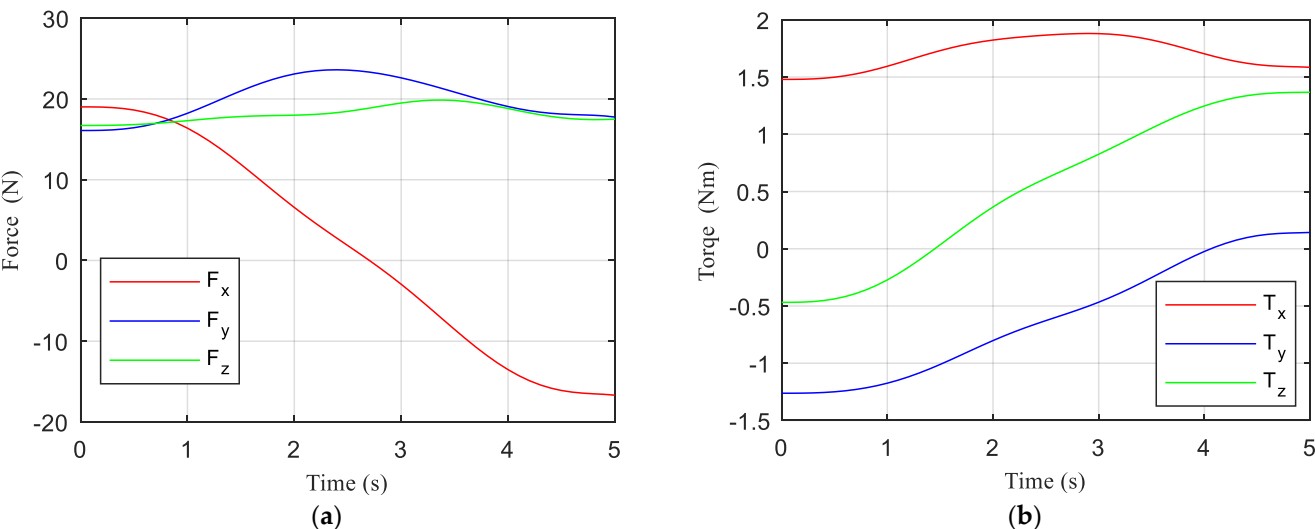

**Figure 6.** The real-time force/torque on the end effector of the humanoid manipulator. (**a**) The real-time force on the end effector; (**b**) The real-time torque on the end effector.

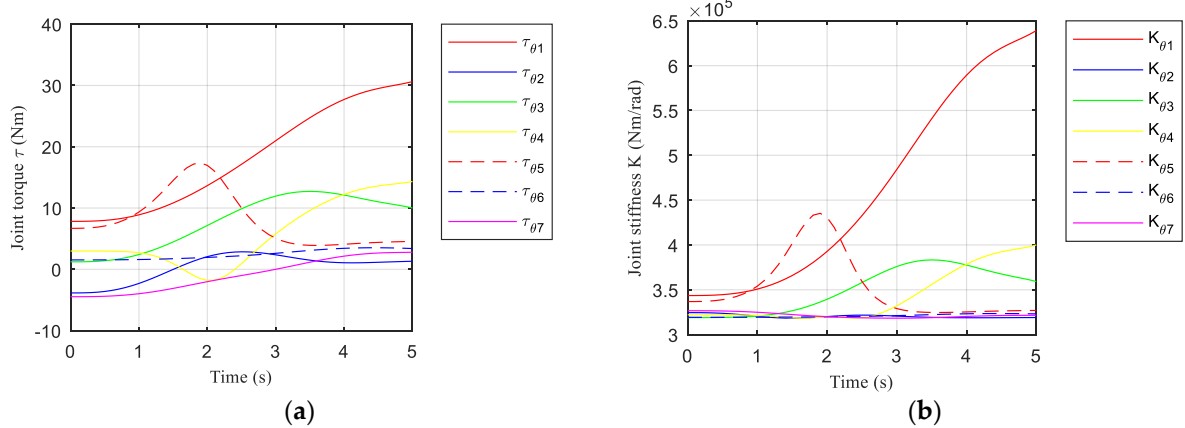

**Figure 7.** The real-time joint torque and joint stiffness of each joint of the humanoid manipulator. (**a**) The real-time joint torque; (**b**) The real-time joint stiffness.

From the above simulation, it can be seen that under the action of gravitational load, due to the position and posture of the humanoid manipulator change in real-time in the process of motion, the force/torque on the end effector of the humanoid manipulator changes in real-time. The real-time variation trend of each joint stiffness is consistent with that of each joint torque during the movement of the humanoid manipulator. However, as the joint torque increases, the real-time variation curve of each joint stiffness becomes steeper, which is beneficial to maintaining the stability of the movement of the humanoid manipulator. By controlling the real-time stiffness of each joint, the generalized displacements of the end effector of the humanoid manipulator can be very small, which does not affect the repeated positioning accuracy of the humanoid manipulator when it works alone.

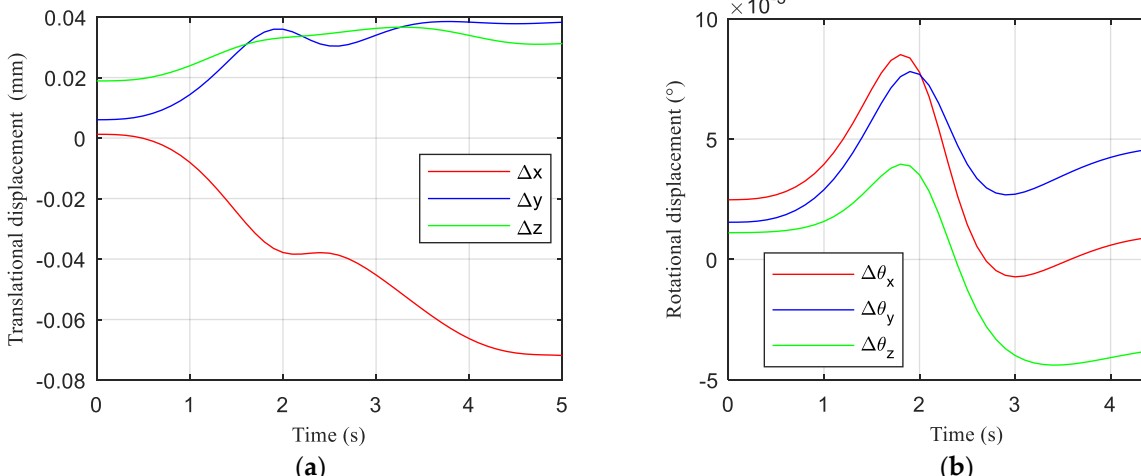

**Figure 8.** The generalized displacements of the end effector of the humanoid manipulator. (**a**) Real-time translational displacements of the end effector; (**b**) Real-time rotational displacements of the end effector.

### 4.2. In the Environment of Human-Robot Interaction

When the humanoid manipulator moves in the environment of human-robot interaction, it may have an accidental collision with the human. In order to reduce injuries caused by accidental collision, it is necessary to adopt the real-time joint stiffness configuration strategy to reduce the injury to human body in collision.

Assuming that the humanoid manipulator is in human-robot interaction environment, human-robot accidental collision may occur at any time, and the collision direction is opposite to the motion direction of the humanoid manipulator. The effective mass of the humanoid manipulator is $m_R = 20$ kg, and considering the weight of the head of the adult human body, the effective mass of the human body is $m_H = 10$ kg. The setting of gravitational load and motion path of the humanoid manipulator are the same as that of the humanoid manipulator working alone. The joint torque $\tau$ and the passive deflection angle $\Delta\theta$ of each joint conform to the hyperbolic tangent relation equation.

According to the above motion planning, the motion speed of the end effector of the humanoid manipulator in the base coordinate system is known in real-time. Since the opposite direction of this velocity is the human-robot collision direction, the collision direction vector in the base coordinate system is calculated as follows:

$$\boldsymbol{n_c} = \left( -\frac{v_{Rx}}{|v_{Rx}|}, -\frac{v_{Ry}}{|v_{Rx}|}, -\frac{v_{Rz}}{|v_{Rx}|} \right) \tag{13}$$

The direction vectors of each axis of the end effector coordinate system of the humanoid manipulator in the base coordinate system are calculated as follows:

$$\begin{cases} \boldsymbol{n_x} = \begin{bmatrix} n_{xx} \\ n_{xy} \\ n_{xz} \end{bmatrix} = {}^0_g\boldsymbol{R} \begin{bmatrix} 1 \\ 0 \\ 0 \end{bmatrix} \\ \boldsymbol{n_y} = \begin{bmatrix} n_{yx} \\ n_{yy} \\ n_{yz} \end{bmatrix} = {}^0_g\boldsymbol{R} \begin{bmatrix} 0 \\ 1 \\ 0 \end{bmatrix} \\ \boldsymbol{n_z} = \begin{bmatrix} n_{zx} \\ n_{zy} \\ n_{zz} \end{bmatrix} = {}^0_g\boldsymbol{R} \begin{bmatrix} 0 \\ 0 \\ 1 \end{bmatrix} \end{cases} \tag{14}$$

The direction cosines of the collision direction vector and each axis of the end effector coordinate system are calculated as follows:

$$n_c \cdot n_x = -\frac{v_{Rx}}{|v_{Rx}|} \cdot n_{xx} - \frac{v_{Ry}}{|v_{Rx}|} \cdot n_{xy} - \frac{v_{Rz}}{|v_{Rx}|} \cdot n_{xz} \tag{15}$$

$$|n_c| = \sqrt{1 + \left(\frac{v_{Ry}}{v_{Rx}}\right)^2 + \left(\frac{v_{Rz}}{v_{Rx}}\right)^2} \tag{16}$$

$$|n_x| = \sqrt{n_{xx}^2 + n_{xy}^2 + n_{xz}^2} \tag{17}$$

$$\cos(n_c, n_x) = \frac{n_c \cdot n_x}{|n_c| \cdot |n_x|} \tag{18}$$

Similarly, direction cosines $\cos(n_c, n_y)$ and $\cos(n_c, n_z)$ can be obtained.

The collision forces on the end effector of the humanoid manipulator in X, Y, and Z directions are calculated as follows:

$$\begin{cases} F_{cx} = F_c \cdot \cos(n_c, n_x) \\ F_{cy} = F_c \cdot \cos(n_c, n_y) \\ F_{cz} = F_c \cdot \cos(n_c, n_z) \end{cases} \tag{19}$$

Considering the gravitational load in real-time motion, the real-time force/torque of the end effector of the humanoid manipulator are calculated as follows:

$$\begin{cases} F_x = F_{cx} + G_x \\ F_y = F_{cy} + G_y \\ F_z = F_{cz} + G_z \\ T_x = T_{gx} \\ T_y = T_{gy} \\ T_z = T_{gz} \end{cases} \tag{20}$$

where $G_x$, $G_y$ and $G_z$ are the components of the gravitational load in the coordinate system of the end effector of the humanoid manipulator; $T_{gx}$, $T_{gy}$, and $T_{gz}$ are respectively the torque exerted by gravitational load on each direction of the end effector of the humanoid manipulator.

According to the joint stiffness configuration strategy, take $A = 0.3$ and $A = 0.5$ respectively, the relevant important indicators during the motion are shown in Table 1.

**Table 1.** The relevant important indicators during the motion.

|  | **Max($v_R$)** | **Max(MSI)** | $T_m$ | $T_c$ | **Max($F_c$)** | $K$ |
|---|---|---|---|---|---|---|
| $A = 0.3$ | 0.553 m/s | 0.040 | 3.15 s | 0.155 s | 149.76 N | $1.1 \times 10^4$N/m |
| $A = 0.5$ | 0.322 m/s | 0.039 | 5.46 s | 0.09 s | 149.91 N | $3.2 \times 10^4$N/m |

It can be seen from Table 1 that the smaller the stiffness of the contact surface during the human-robot collision, the longer the collision time $T_c$, and the greater the movement speed of the humanoid manipulator that can be allowed to ensure that the human body is not injured.

It is assumed that the stiffness of the human body surface is $K_H = 50,000$N/m, if the comprehensive surface contact stiffness $K$ between the humanoid manipulator and human body is known, then the surface stiffness $K_R$ of the humanoid manipulator in the collision direction is calculated as follows:

$$K_R = \frac{K_H - K}{K_H K} \tag{21}$$

The parameter matrices $a$ and $b$ in Equation (6) need to be reasonable. Assuming that $a_i = \frac{50}{\pi}$, $i = 1 \cdots 7$, we obtain the parameter matrix $b$ according to the surface contact stiffness $K_R$ of the humanoid manipulator. As shown in Table 2, if the constant $A$ is different, the surface contact stiffness $K_R$ of the humanoid manipulator and the force $F_y$ of the end effector in Y direction are different. $\Delta y'$ is the expected translational displacement of the end effector in Y direction, we try to obtain the reasonable parameter matrix $b$ through trial and error method, then the actual translational displacement of the end effector in Y direction is $\Delta y \approx \Delta y'$.

**Table 2.** Generalized displacements of the end effector under different parameters.

|  | $K_R$ | $F_y$ | $\Delta y'$ | $b_i(i=1\cdots 7)$ | $\Delta y$ |
|---|---|---|---|---|---|
| $A = 0.2$ | $5.3 \times 10^3 \text{N/m}$ | $-16.3$ N | $-3.06$ mm | $2.4 \times 10^{-3}$ | $-3.1$ mm |
| $A = 0.3$ | $1.4 \times 10^4 \text{N/m}$ | $-19.9$ N | $-1.42$ mm | $1.1 \times 10^{-3}$ | $-1.5$ mm |
| $A = 0.4$ | $3.4 \times 10^4 \text{N/m}$ | $-21.1$ N | $-0.62$ mm | $5 \times 10^{-4}$ | $-0.66$ mm |
| $A = 0.5$ | $9.3 \times 10^4 \text{N/m}$ | $-20.8$ N | $-0.22$ mm | $1.7 \times 10^{-4}$ | $-0.23$ mm |
| $A = 0.6$ | $2.6 \times 10^6 \text{N/m}$ | $-19.39$ N | $-0.08$ mm | $6 \times 10^{-5}$ | $-0.08$ mm |

According to Table 2, in the human-robot interaction environment, we set the parameter matrix $b = \left[8 \times 10^{-4}, 8 \times 10^{-4}, 8 \times 10^{-4}, 8 \times 10^{-4}, 8 \times 10^{-4}, 8 \times 10^{-4}, 8 \times 10^{-4}\right]^T$. Take $A = 0.35$ and $T_m = 4s$, the real-time situation of the joint space stiffness, force/torque on the end effector, MSI etc. of the humanoid manipulator are shown in Figures 9–12 respectively.

It can be seen from Figures 9 and 10 that after adopting the real-time joint stiffness configuration strategy, the force on the end effector is controlled within a certain range, and the collision force will not cause harm to the human body. Meanwhile, the value of real-time MSI is far less than $MSI_{max}$.

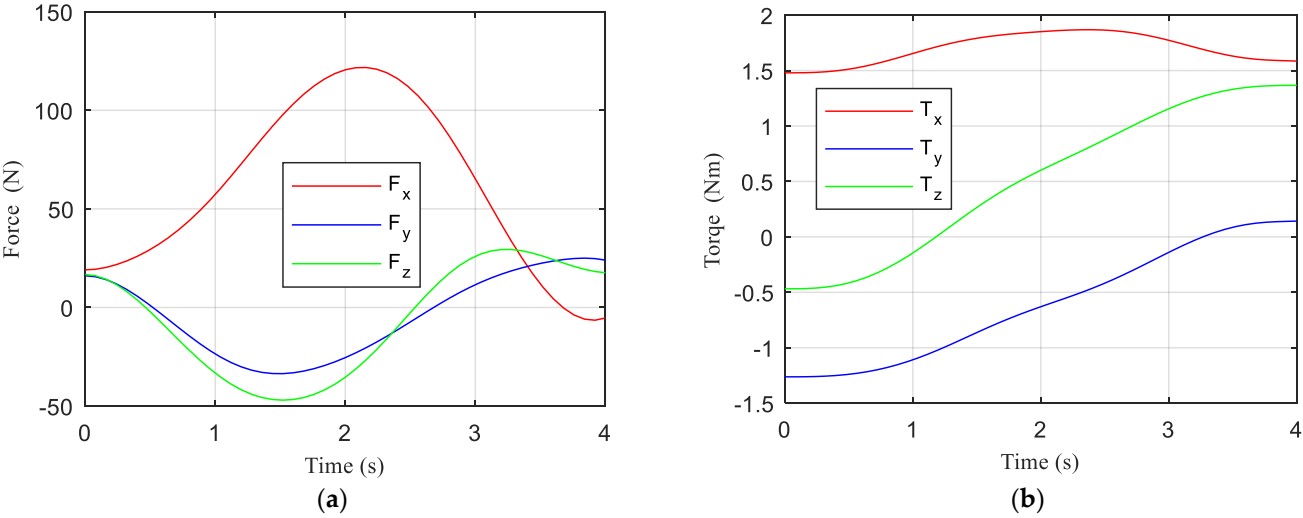

**Figure 9.** The real-time force/torque on the end effector of the humanoid manipulator. (**a**) Real-time force on the end effector; (**b**) Real-time torque on the end effector.

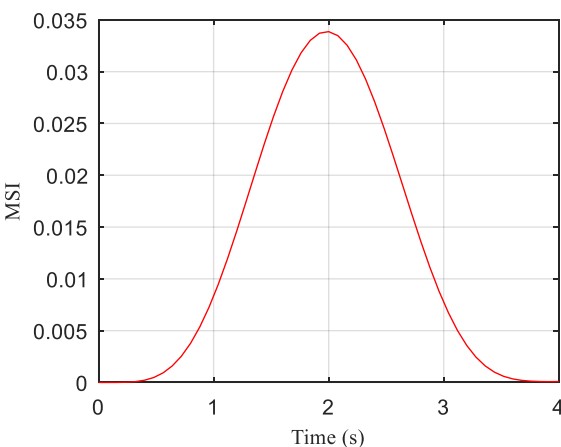

**Figure 10.** The real-time MSI of the humanoid manipulator.

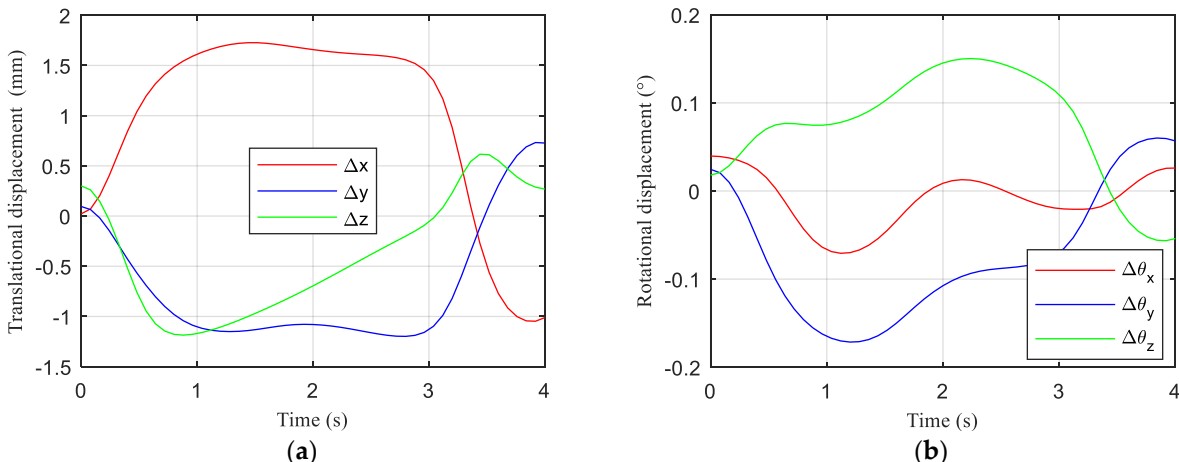

**Figure 11.** The generalized displacements of the end effector of the humanoid manipulator. (**a**) Real-time translational displacements of the end effector; (**b**) Real-time rotational displacements of the end effector.

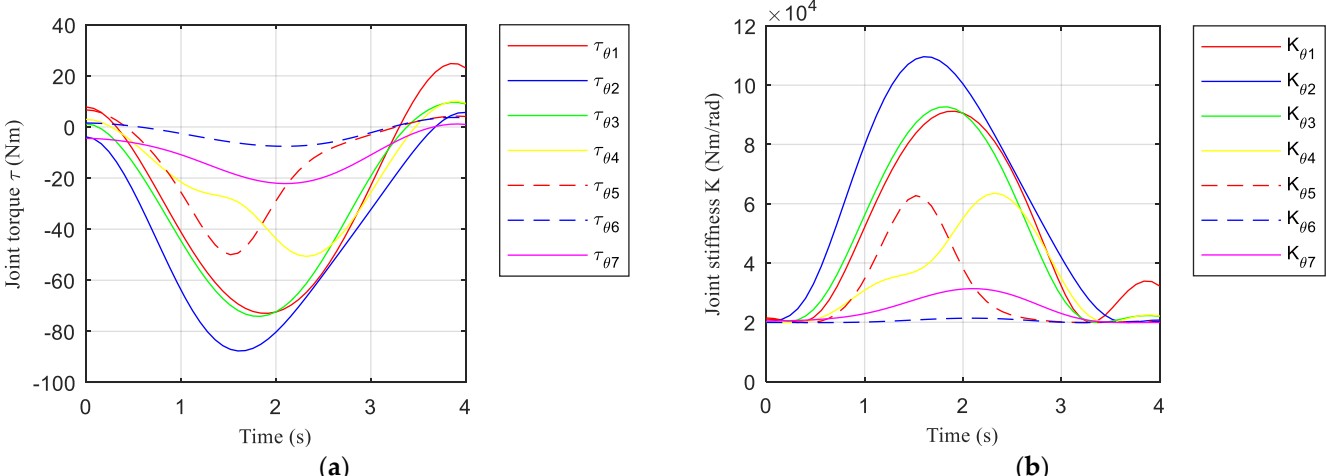

**Figure 12.** The real-time joint torque and joint stiffness of each joint of the humanoid manipulator. (**a**) The real-time joint torque; (**b**) The real-time joint stiffness.

As can be seen from Figures 11 and 12, compared with the humanoid manipulator working alone, in the human-robot interaction environment, the stiffness of each joint of the

humanoid manipulator is smaller, and the overall flexibility of the humanoid manipulator is better, which means that the human body is safer in a human-robot collision.

It is assumed that if there is no real-time stiffness configuration strategy, then the surface contact stiffness of the robot is set as $K_R = 10^5$ N/m. According to Equations (1) and (3), in case of accidental collision, the comparison of results with and without real-time joint stiffness configuration strategy is shown in Table 3.

**Table 3.** Comparison of results with and without real-time joint stiffness configuration strategy.

|  | **Max($F_c$)** | **Max(MSI)** |
|---|---|---|
| With stiffness strategy | 138.7N | 0.034 |
| Without stiffness strategy | 205.7N | 0.085 |

It can be seen from Table 3 that the real-time joint stiffness configuration strategy can improve human safety in human-robot collision.

In practical applications, such as robot grasping or placing tasks, the lower joint stiffness of the robot will affect its position accuracy. Therefore, a position compensation method is needed to improve the accuracy of the robot.

Suppose that the desired trajectory of the robot is $P_e$, the flexible deformation of the end effector due to the low joint stiffness of the robot is $\delta P$, and the compensation for the trajectory is $\delta P_c$, then the real trajectory of the robot is calculated as follows:

$$P_r = P_e + \delta P - \delta P_c \tag{22}$$

According to the above desired trajectory, the real-time position error without compensation is shown in Figure 13a, and the position error after trajectory compensation is shown in Figure 13b.

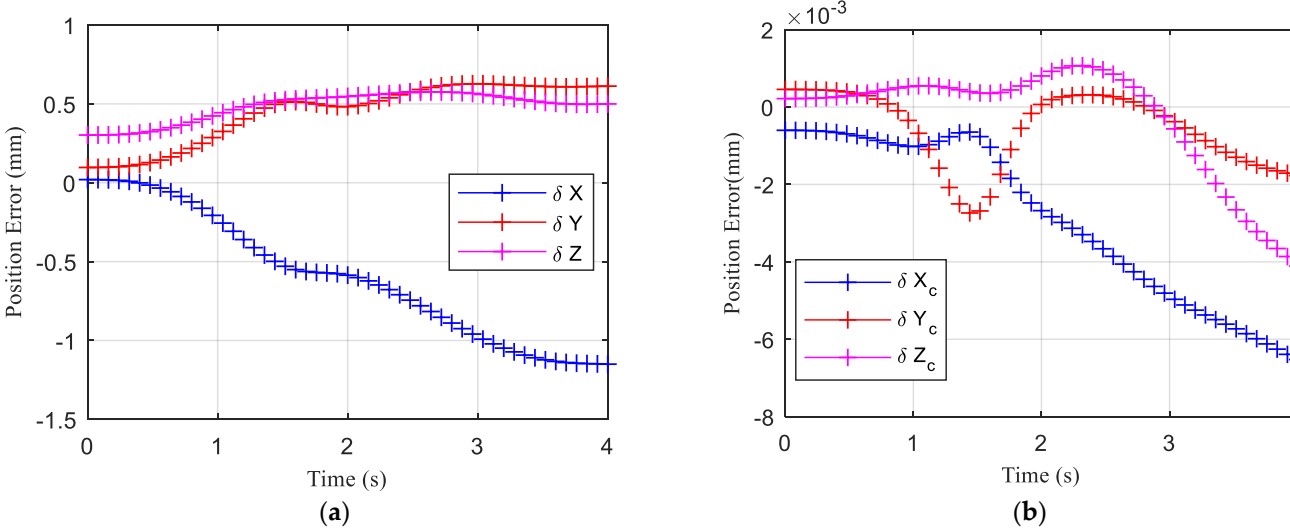

**Figure 13.** The real-time position error of the humanoid manipulator in motion. (**a**) No trajectory compensation; (**b**) With trajectory compensation.

It can be seen from Figure 13 that the position error of the end effector after trajectory compensation is less than 0.1mm, which does not affect the robot's grasping and placement tasks.

In addition, in the application of the joint stiffness configuration strategy, if the robot is mobile, the speed of the robot itself and the speed of the mobile car should be considered simultaneously in the human-robot collision.

## 5. Conclusions

In this paper, the real-time joint stiffness configuration strategy of a series of parallel hybrid 7-DOF humanoid manipulators with flexible joints in continuous motion is studied. We find that the change of the joint space stiffness or the posture of the humanoid manipulator can directly affect the Cartesian space stiffness of the humanoid manipulator. The hyperbolic tangent relation equation between the joint torque and the passive joint deflection angle of the humanoid manipulator is proposed, which is beneficial for real-time calculation of joint stiffness and obtaining reasonable joint stiffness. According to the working environment of the humanoid manipulator, the joint stiffness configuration strategy of the humanoid manipulator in continuous motion is given. When the humanoid manipulator works alone, the joint space stiffness will be larger to ensure the working accuracy of the humanoid manipulator in Cartesian space. When the humanoid manipulator works in the human-robot interaction environment, in order to prevent the human body from being injured in human robot accidental collision, we consider the manipulator safety index and human injury threshold, and the motion speed and joint stiffness of the humanoid manipulator are optimized in advance. The simulation results show that the joint stiffness configuration strategy can effectively improve the safety of the human body in human-robot collision, and different parameters will affect the flexibility of the humanoid manipulator. In addition, in application, when the joint space stiffness of the robot is lower, the position accuracy can be improved by trajectory compensation, and the position error of the end effector after trajectory compensation is less than 0.1 mm.

In the future, we will focus on the relationship between anthropomorphic motion and joint space stiffness of the humanoid manipulator. In addition, the application and improvement of the joint stiffness configuration strategy is our concern. We hope that the strategy can be applied to mobile as well as static robots.

**Author Contributions:** Funding acquisition, S.W.; Methodology, Y.Y.; Supervision, H.S.; Writing—original draft, Y.Y.; Validation, Y.Z.; Writing—review & editing, S.W., H.S. and Y.Z. All authors have read and approved the final paper. All authors have read and agreed to the published version of the manuscript.

**Funding:** This work was supported by the Beijing Municipal Natural Science Foundation under Grant L172031.

**Institutional Review Board Statement:** Not applicable.

**Informed Consent Statement:** Not applicable.

**Data Availability Statement:** Not applicable.

**Conflicts of Interest:** The authors declare that there are no conflicts of interest.

## Appendix A. Structure Parameters of the Humanoid Manipulator

The mechanism diagram of the humanoid manipulator is shown in Figure A1.

Based on the research on the range of motion of each joint of the human arm [31–33], the posture angle ranges of each motion platform of the humanoid manipulator are shown in Table A1.

**Table A1.** The posture angle range of each motion platform of the humanoid manipulator.

| Posture Angle | Range of Motion (°) |
| --- | --- |
| Flexion and extension angle of shoulder joint $\beta_{s1}$ | $[-160, 40]$ |
| Abduction and adduction angle of shoulder joint $\gamma_{s2}$ | $[-90, 30]$ |
| External and internal rotation angle of shoulder joint $\alpha_{s2}$ | $[-70, 70]$ |
| Flexion and extension angle of elbow joint $\beta_e$ | $[-150, 10]$ |
| Pronation and supination angle of elbow joint $\alpha_e$ | $[-90, 90]$ |
| Radial deviation and ulnar deviation of wrist joint $\gamma_w$ | $[-40, 40]$ |
| Dorsal extension and palmar flexion of wrist joint $\beta_w$ | $[-60, 60]$ |

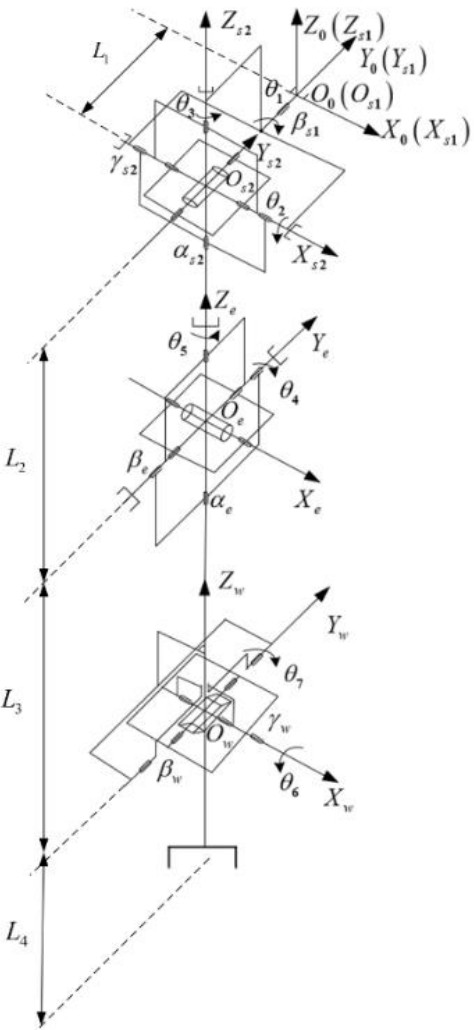

**Figure A1.** The mechanism diagram of the humanoid manipulator.

By calculation, the input angle range of each joint of the humanoid manipulator are shown in Table A2.

**Table A2.** The input angle range of each joint of the humanoid manipulator.

| The Input Angle of Each Joint | Range of Motion (°) |
|---|---|
| Input angle of shoulder joint $\theta_1$ | $[-160, 40]$ |
| Input angle of shoulder joint $\theta_2$ | $[-90, 30]$ |
| Input angle of shoulder joint $\theta_3$ | $[-90, 90]$ |
| Input angle of elbow joint $\theta_4$ | $[-150, 10]$ |
| Input angle of elbow joint $\theta_5$ | $[-90, 90]$ |
| Input angle of wrist joint $\theta_6$ | $[-40, 40]$ |
| Input angle of wrist joint $\theta_7$ | $[-70, 70]$ |

As shown in Figure A1, the size of the humanoid manipulator is given as follows.

$$\begin{cases} L_1 = 0.18m \\ L_2 = 0.626m \\ L_3 = 0.51m \\ L_4 = 0.16m \end{cases} \tag{A1}$$

According to the forward kinematics, the workspace of the end effector of the humanoid manipulator is shown in Figure A2.

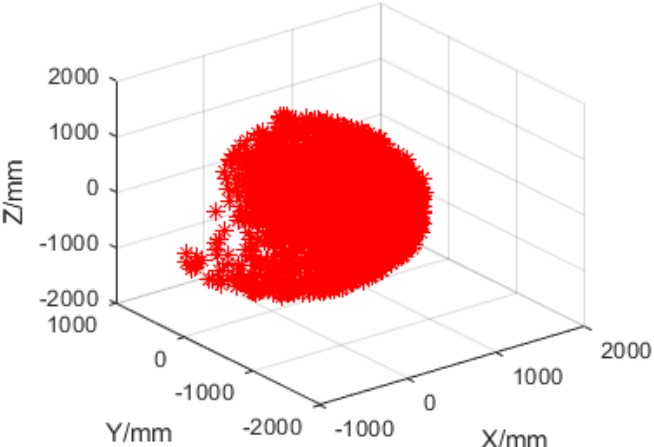

**Figure A2.** The workspace of the end effector of the humanoid manipulator.

### Appendix B. Calculation of Gravitational Load

Assuming that the load mass of the humanoid manipulator is $m_{load}$, then the gravity vector of the load in the base coordinate system of the humanoid manipulator is $^0P = [0, 0, -m_{load}g]^T$. The gravity vector of the load in the coordinate system of the end effector is calculated as follows:

$$\begin{bmatrix} G_x \\ G_y \\ G_z \end{bmatrix} = {}^g_0R \begin{bmatrix} 0 \\ 0 \\ -m_{load}g \end{bmatrix} \tag{A2}$$

where $G_x$, $G_y$ and $G_z$ are the components of the gravitational load in the coordinate system of the end effector of the humanoid manipulator; ${}^g_0R$ is the transformation matrix of base coordinate system relative to the humanoid manipulator end effector coordinate system.

The coordinate of the center of gravity of the load relative to the coordinate system of the end effector of the humanoid manipulator is $(x_G, y_G, z_G)$, the torque exerted by gravitational load on the end effector of the humanoid manipulator is shown in Figure A3.

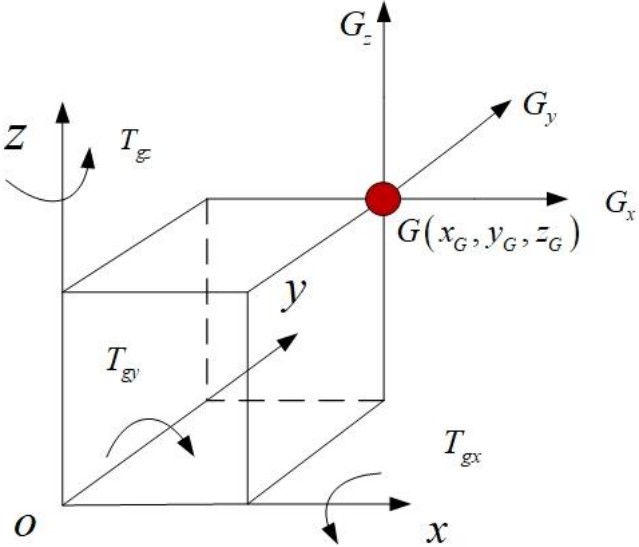

**Figure A3.** The torque exerted by gravitational load on the end effector of the humanoid manipulator.

According to Figure A3, the calculation is as follows:

$$\begin{cases} T_{gx} = G_z y_G - G_y z_G \\ T_{gy} = G_x z_G - G_z x_G \\ T_{gz} = G_y x_G - G_x y_G \end{cases} \tag{A3}$$

where $T_{gx}$, $T_{gy}$ and $T_{gz}$ are respectively the torque exerted by gravitational load on each direction of the end effector of the humanoid manipulator.

### Appendix C. Motion Planning of the Humanoid Manipulator

The starting point and end point of the end effector of the humanoid manipulator are $P_A = (300, -500, 700, 30, -100, 160)$ and $P_B = (900, 100, 300, 40, -80, 120)$, and the units of position and Euler angle are millimeter and degree respectively. The joint angle and real-time trajectory of the end effector of the humanoid manipulator in real-time motion are shown in Figures A4 and A5.

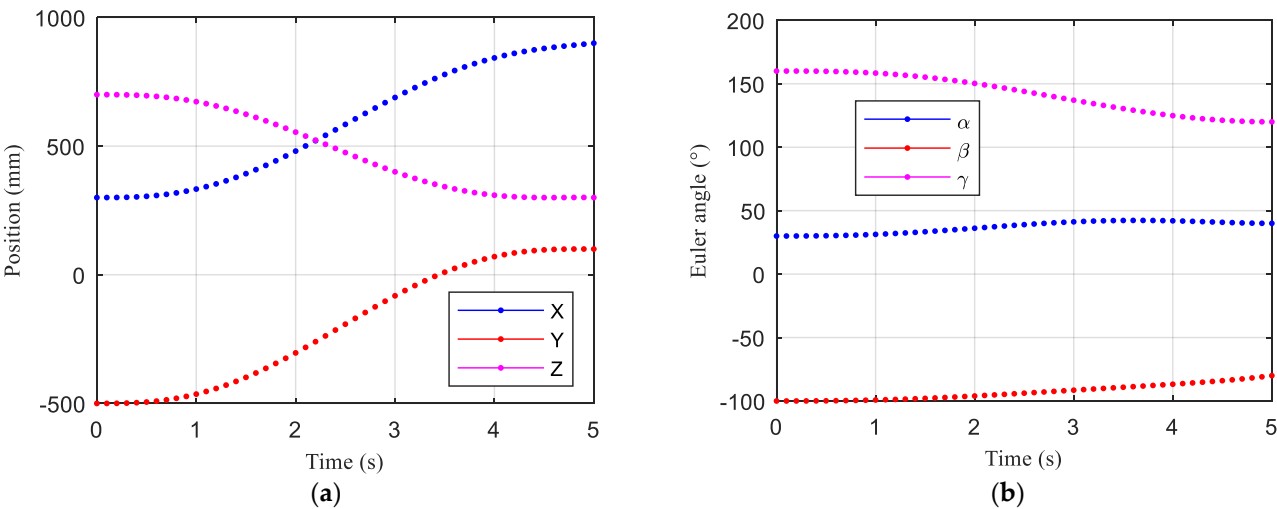

**Figure A4.** The real-time trajectory of the end effector of the humanoid manipulator. (**a**) The real-time position of the end effector; (**b**) The real-time Euler angle of the end effector.

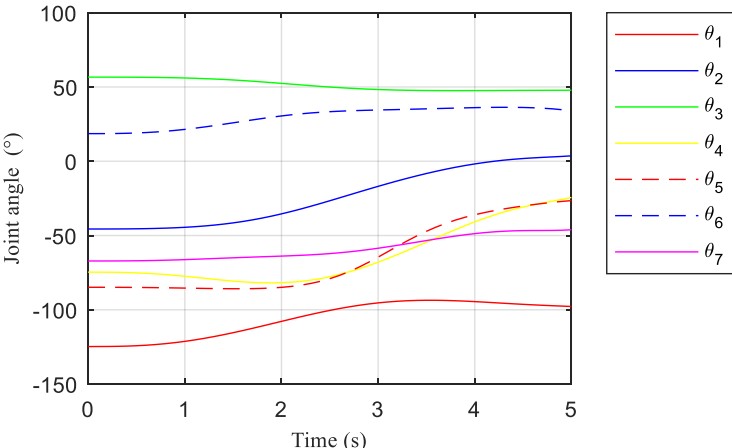

**Figure A5.** The real-time angles of the joints of the humanoid manipulator during the movement.

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
