# Peer review of "Research on Real-Time Joint Stiffness Configuration of a Series Parallel Hybrid 7-DOF Humanoid Manipulator in Continuous Motion"

_applsci, doi:10.3390/app11052433_

Round 1

Reviewer 1 Report

The paper presented describes a study on joint stiffness variations of a humanoid manipulator depending on the working area – either the robot works alone or in collaboration with humans. Human robot interaction environments will become more and more important resulting in the development of new and reliable safety concepts. The paper is well written, the figures are clear and excellent readable and the topic is related to a very important field in robotics.

In the reviewers opinion, there are some revisions required related to the practical use of the strategy presented in this paper. To support the performance of the strategy in real-time joint stiffness configuration application-related results need to be added.

Comments to the individual sections

1. Introduction

The introduction is well written and gives a good overview of the topic and the importance of joint stiffness configuration depending on the robots working environment.

2. Mechanism and Characteristics of the Humanoid Manipulator

This section describes detailed the humanoid manipulator with 7 DOF used in the study presented. It is not really clear, was the humanoid manipulator a model only in simulation or a real robotic system. It would be interesting, why this model was selected for the study instead a commercially available humanoid robot.

3. The Joint Space Stiffness and the Cartesian Stiffness

In this section, the joint space stiffness and the Cartesian stiffness are explained and their relationship between. Figures and equations support the understanding.

4. Real-time Joint Stiffness Configuration Strategy

Section 4 forms the main part of the paper and presents the strategy for real time joint stiffness configuration in detail. It would be interesting, if this strategy was applied to a real humanoid robotic system. Furthermore, results for different robot-human collision scenarios would support the performance of the strategy developed. It would be also interesting, if this strategy can be applied for both stationary and mobile robots. Can the authors provide a statement for this?

One question: In line 423, the effective mass of a humanoid manipulator was set to 20 kg and the effective mass of a human body was set to 10 kg. Can the authors explain this?

5. Conclusions

The last chapter summarizes the aim and the scope of the study. It was mentioned, that while working the robot alone, the joint space stiffness is larger to ensure a high accuracy. And in human-robot interaction environments, the joint space stiffness is lower to ensure maximum safety of the humans. How much the accuracy (position accuracy?) is reduced in such an environment? Can the robot fulfil tasks such as pick and place? I mean, what is the loss of accuracy with lower joint space stiffness?

Reviewer 2 Report

The manuscript describes an interesting method to allow interaction safety while interacting with humans with a robotic arm. 

The content of the paper is not well organised and, consequently, the presentation is not clear.   I would suggest starting from the explanation of the proposed algorithms before describing the components. 

The paper body should focus on stiffness adjustment and the safety metrics needed to deliver the paper's central message. Thus, it would be best to consider moving the redundancy approach, gravity formulation and the planner in the appendix. 

Round 2

Reviewer 2 Report

The authors addressed my comments adequately